# Discovery, Genomic Sequence Characterization and Phylogenetic Analysis of Novel RNA Viruses in the Turfgrass Pathogenic *Colletotrichum* spp. in Japan

**DOI:** 10.3390/v14112572

**Published:** 2022-11-20

**Authors:** Islam Hamim, Syun-ichi Urayama, Osamu Netsu, Akemi Tanaka, Tsutomu Arie, Hiromitsu Moriyama, Ken Komatsu

**Affiliations:** 1Laboratory of Plant Pathology, Graduate School of Agriculture, Tokyo University of Agriculture and Technology (TUAT), Fuchu 183-8509, Tokyo, Japan; 2International Research Fellow, Japan Society for the Promotion of Science, Chiyoda 102-0083, Tokyo, Japan; 3Department of Plant Pathology, Bangladesh Agricultural University, Mymensingh 2202, Bangladesh; 4Laboratory of Fungal Interaction and Molecular Biology (Donated by IFO), Department of Life and Environmental Sciences, University of Tsukuba, Tsukuba 305-8577, Ibaraki, Japan; 5Microbiology Research Center for Sustainability (MiCS), University of Tsukuba, Tsukuba 305-8577, Ibaraki, Japan; 6Product Development Section, Development Division, MARUWA BIOCHEMICAL Co., Ltd., Inashiki-Gun, Tsukuba 300-1161, Ibaraki, Japan; 7Laboratory of Molecular and Cellular Biology, Faculty of Agriculture, Tokyo University of Agriculture and Technology (TUAT), Fuchu 183-8509, Tokyo, Japan

**Keywords:** RNA virus, viral genome, FLDS, RdRp, *Colletotrichum* strains

## Abstract

Turfgrass used in various areas of the golf course has been found to present anthracnose disease, which is caused by *Colletotrichum* spp. To obtain potential biological agents, we identified four novel RNA viruses and obtained full-length viral genomes from turfgrass pathogenic *Colletotrichum* spp. in Japan. We characterized two novel dsRNA partitiviruses: Colletotrichum associated partitivirus 1 (CaPV1) and Colletotrichum associated partitivirus 2 (CaPV2), as well as two negative single-stranded (ss) RNA viruses: Colletotrichum associated negative-stranded RNA virus 1 (CaNSRV1) and Colletotrichum associated negative-stranded RNA virus 2 (CaNSRV2). Using specific RT-PCR assays, we confirmed the presence of CaPV1, CaPV2 and CaNSRV1 in dsRNAs from original and sub-isolates of *Colletotrichum* sp. MBCT-264, as well as CaNSRV2 in dsRNAs from original and sub-isolates of *Colletotrichum* sp. MBCT-288. This is the first time mycoviruses have been discovered in turfgrass pathogenic *Colletotrichum* spp. in Japan. CaPV1 and CaPV2 are new members of the newly proposed genus “Zetapartitivirus” and genus *Alphapartitivirus*, respectively, in the family *Partitiviridae*, according to genomic characterization and phylogenetic analysis. Negative sense ssRNA viruses CaNSRV1 and CaNSRV2, on the other hand, are new members of the family *Phenuiviridae* and the proposed family “Mycoaspirividae”, respectively. These findings reveal previously unknown RNA virus diversity and evolution in turfgrass pathogenic *Colletotrichum* spp.

## 1. Introduction

Colletotrichum is an economically valuable fungal genus that affects a broad range of hosts, including agricultural crops [1]. Especially, anthracnose disease epidemics have affected turfgrass on golf courses used for landscaping and putting greens, resulting in significant economic losses and an unwelcome but necessary spike in fungicide use [2]. Anthracnose is caused by *Colletotrichum* spp., which induces foliar blight, basal stem rot and eventually host mortality in turfgrass species. These symptoms progress to irregular patches. Because of the importance of anthracnose disease, researchers have been motivated to conduct research on the detection, characterization and management of *Colletotrichum* spp. in turfgrass hosts [3]. However, care should be taken to use a chemical strategy to manage this fungal disease in some cases, due to fungicide resistance and environmental pollution. Given these challenges, biological control agents are attractive solutions for disease management due to their environmental safety.

Fungal viruses, also called mycoviruses, are widespread in fungi. There have also been multiple reports of mycovirus infections in *Colletotrichum* spp., mostly caused by partitiviruses. Mycoviruses from the *Chrysoviridae* family as well as an ourmia-like virus have also been reported in *Colletotrichum* spp. [3,4]. To date, mycoviruses associated with *Colletotrichum* spp. have contained plus-strand (+) single-stranded RNA (ssRNA) or double-stranded RNA (dsRNA) genomes, while no mycovirus with a (−) ssRNA or DNA genome has been identified. Mycoviruses primarily induce latent infections in fungal hosts, while there have been some reports on the virulence of mycoviruses in their hosts. Mycovirus infection can cause significant physiological changes to a fungus, such as abnormal morphological traits, decreased mycelial growth and inhibited spore generation, in addition to its impact on pathogenicity in plants [5]. Because mycoviruses that can affect fungal physiology would ultimately lead to the complete loss of or reduction in the virulence of certain fungal pathogens such as *Colletotrichum* spp., research on mycoviruses that infect plant-pathogenic fungi can contribute to the development of substitutes to chemical-based disease management, perhaps resulting in less harm to humans and the environment. Many researchers have been inspired to hunt mycoviruses for useful biocontrol agents in recent decades owing to the potential biological usefulness of mycovirus-mediated hypovirulence to control fungal diseases, as demonstrated by the successful use of Cryphonectria hypo-virus 1 to control chestnut blight disease in Europe [5,6,7]. Furthermore, several mycoviruses can increase the virulence of the host fungus, which might be utilized to characterize the molecular mechanism of virulence modulation in fungi [8,9]. Besides these, the discovery of mycoviruses is expanding our understanding of virus diversity, ecology and evolution.

Generally, dsRNA has been employed as an indicator of mycovirus infection. The combination of a quick and selective extraction technique for dsRNA from fungal cells and the conventional agarose gel electrophoresis has accelerated large-scale mycovirus screening [10,11,12,13]. However, because of the relatively low sensitivity of agarose gel electrophoresis, it cannot identify low-titer infections of mycoviruses. Over the last decade, the development of metagenomic and meta-transcriptomic analyses using deep-sequencing, which is more sensitive to detect small amounts of nucleic acids, has allowed increasing numbers of novel viruses to be identified in ecologically variable environments [14,15]. Indeed, Illumina sequencing technology has effectively discovered mycoviruses from fungal isolates previously thought to be virus-free [16].

Deep-sequencing approaches such as total RNA sequencing, on the other hand, have some limitations. One of these limitations is loss of information on the terminal sequences of a novel or unknown viral genome due to the absence of a reference genome. This would lead to difficulty in the identification of all cognate segments of one virus. Indeed, some mycoviruses have multi-segmented RNA genomes, of which one segment has an open reading frame (ORF) that encodes an RNA-dependent RNA polymerase (RdRp), which is necessary and ubiquitous in RNA viruses. In most cases, although the conserved sequences at their terminal ends may be used to identify cognate segments, the technological constraints mentioned above have hampered the finding of cognate segments and new viral sequences. To address this issue, we employed the ‘Fragmented and primer-Ligated dsRNA Sequencing (FLDS)’ technique [17,18]. This approach generates trustworthy terminal sequences for each genome, allowing segmented RNA genomes of viruses to be identified in a homology-independent manner, which can be used to discover and determine the full-length genome sequences of novel mycoviruses.

To date, it remains elusive whether mycoviruses can infect turfgrass pathogenic *Colletotrichum* spp. In this paper, we report the detection, genomic sequence characterization and phylogenetic analyses of two novel dsRNA viruses and a (−) ssRNA virus from turfgrass pathogenic *Colletotrichum* sp. MBCT-264, as well as a (−) ssRNA virus from turfgrass pathogenic *Colletotrichum* sp. MBCT-288 in Japan. This is the first discovery of mycoviruses from turfgrass pathogenic *Colletotrichum* sp. in Japan. Findings from this study will contribute to our understanding of mycoviral diversity, ecology and evolution, as well as provide new insight into the search for viable biocontrol agents to manage turfgrass pathogenic *Colletotrichum* spp.

## 2. Material and Methods

### 2.1. Turfgrass Pathogenic Colletotrichum Strains and Culture Conditions

In the autumn of 2019 and spring of 2020, nine *Colletotrichum* isolates were obtained from diseased turfgrasses in different locations in Japan by using potato dextrose agar (PDA) with 0.5 µg/mL sodium ampicillin and 0.01 µg/mL rifampicin. The isolates were transferred to a slant PDA and stored at 10 °C until use (Table 1). The identification of *Colletotrichum* spp. was based on morphological characteristics [19] and phylogenetic analysis of ITS (internal transcribed spacer), Sod2, Apn2 and Apn2/Mat1 gene segments [20].

### 2.2. Extraction and Purification of dsRNA

The mycelial blocks of *Colletotrichum* strains were transferred from PDA plates to potato sucrose broth and cultured for seven days at 27 °C with an orbital shaker at 120 rpm. Collected mycelia were dried and kept at −30 °C before use. Total nucleic acids were isolated and dsRNAs were purified using Cellulose D (Advantec Toyo Roshi Kaisha, Ltd., Tokyo, Japan) as described by Okada et al. [11]. Total nucleic acids were extracted with equal volumes of phenol-chloroform-isoamyl alcohol (PCI; 25:24:1) after crushing 0.1 g (dry weight) of fungal mycelium in 0.5 mL of extraction buffer (100 mM NaCl, 10 mM Tris-HCl pH 8.0, 1 mM EDTA, 1% SDS and 0.1% (*v*/*v*) mercapto-ethanol). After centrifugation, the supernatant was combined with ethanol (final 16%) and dsRNA was purified by employing a spin column. Finally, the dsRNAs were precipitated with ethanol and preserved at −80 °C. The isolated dsRNAs were visualized using electrophoresis in 0.8% (*w*/*v*) agarose gels stained with ethidium bromide. 

### 2.3. dsRNA Sequencing to Identify Mycovirus Genome

The dsRNA samples from *Colletotrichum* sp. strains MBCT-264 and MBCT-288 were prepared individually for viral genome sequencing by the “fragmented and primer ligated dsRNA sequencing (FLDS)” method [18]. Briefly, twenty nanograms of dsRNA were fragmented using a Covaris S220 ultra-sonicator (Woburn, MA, USA). The fragmentation conditions were as follows: 40 s run duration, 50.0 W peak power, 2.0 percent duty factor and 200 cycles per burst. A U2 adaptor (final conc. 0.4 μM) was ligated to the fragmented dsRNA using T4 RNA ligase (final conc. 0.8 U/μL) (Takara Bio Inc., Kusatsu, Japan). After denaturing the ligated dsRNA product, single-stranded cDNA (sscDNA) was generated with a U2-complementary primer employing the SMARTer RACE 5′/3′ Kit (Takara). dscDNA was generated via PCR employing a blend of a U2-complementary primer and a universal primer (supplied by the SMARTer RACE 5′/3′ Kit). For Illumina sequencing, cDNA libraries were created and the Illumina MiSeq platform was used to determine the 300 bp paired-end sequences from each fragment (Illumina, San Diego, CA, USA).

### 2.4. Analysis of FLDS-Generated Sequence Reads

Clean reads were obtained before identifying RNA viruses by eliminating low-quality reads and adaptor sequences, as well as contaminated rRNA reads [21]. The cleaned reads were submitted to de novo assembly employing the CLC Genomics Workbench version 11.0 (CLC Bio, Aarhus, Denmark). The generated assemblies were analyzed using the assembly Tablet viewer [22]. We defined the terminal end of an RNA genome of viruses when the terminal end of a contig ended with the same nucleotides for more than 10 reads or a polyA sequence was present [17]. If a contig contained termini at both ends, it was considered a full RNA genome sequence. The terminal sequence identities of the segments were used to examine the presence of multi-segment genomes of a virus [23]. The Basic Local Alignment Search Tool (BLAST) was applied to identify similarities among known nucleotide or protein sequences. The BLASTn and BLASTx applications were employed to determine the sequence identities of FLDS-detected genome sequences by comparing them to existing nucleotide or protein sequences in the NCBI database [24]. The RdRp-encoding segments with >90% nucleic acid sequence homologies were designated as single operational taxonomic units (OTUs) based on Chiba’s criteria [23]. The distinct and new OTUs were considered as novel viral species and were thus provisionally named based on the taxonomic linage of the top ‘hits’ for viruses in the BlastX investigation against the non-redundant (nr) protein sequences database of NCBI.

### 2.5. Validation of Genome Segments of Mycoviruses by RT-PCR

The presence of putative mycoviral genome segments in original fungal dsRNA samples was confirmed by reverse transcription-PCR (RT-PCR) using specific primer sets designed based on the de novo assembled contigs generated from FLDS reads (Appendix A). RT-PCR assays were performed using the PrimeScript^TM^ One Step RT-PCR Kit Ver. 2 (Takara, Japan). For each RT-PCR reaction, about five nanograms of heat-denatured dsRNA were used. Reaction mixtures were incubated at 50 °C for 30 min and 94 °C for 2 min, followed by 30 cycles of 94 °C for 30 s, 55 °C or 56 °C for 30 s and 72 °C for 1 min, with a final extension of 72 °C for 5 min. PCR products were run on the 1% (*w*/*v*) agarose gel and the amplified fragments were excised, purified using the FastGene Gel/PCR Extraction Kit (Nippon Genetics, Tokyo, Japan) and then used for direct Sanger sequencing (Eurofins Genomics Co., Ltd., Tokyo, Japan). For reproducibility, dsRNA preparations from isogenic isolates of fungal strains were also used as a template for RT-PCR.

### 2.6. Analysis of Genomes of Identified Viruses

Open reading frames (ORFs) in each of the full-length genome sequences were determined using the ORF Finder program (http://www.ncbi.nlm.nih.gov/gorf/gorf.html, accessed on 29 August 2021) with the “standard” genetic code. Molecular weights of the predicted proteins were calculated using the Sequence Manipulation Suite (https://bioinformatics.org/sms/, accessed on 20 June 2022). The putative function of the predicted protein was estimated by database searches of the full-length genome sequences of mycoviruses or their deduced polypeptides using the programs BLASTn and BLASTp, respectively. The CDD database (http://www.ncbi.nlm.nih.gov/Structure/cdd/wrpsb.cgi, accessed on 27 March 2022), the Pfam database (http://pfam.sanger.ac.uk/, accessed on 27 March 2022) and the PROSITE database (http://www.expasy.ch/, accessed on 27 March 2022) were used to explore motifs contained in the deduced polypeptide sequences. CLUSTALW and MUSCLE were used to make multiple alignments [25,26]. Mfold (http://www.unafold.org/mfold/applications/rna-folding-form.php, accessed on 2 June 2022) predicted secondary architectures of terminal sequences of mycovirus genomes [27]. Pairwise similarities between the genome sequences and amino acid sequences were determined using the Clustal Omega online tool (https://www.ebi.ac.uk/Tools/msa/clustalo/ accessed on 22 July 2022) with default parameters.

### 2.7. Phylogenetic Analysis of Identified Viruses

RdRp or coat protein (CP) sequences of all identified viruses and closest homologues from the NCBI database were aligned using the Clustal Omega or MUSCLE implemented in MEGAX with default parameters [28]. On the basis of the aligned amino acid sequences, the maximum-likelihood trees were computed by MEGAX [29] or IQ-TREE software [30] with default parameters. The best substitution model, which is shown in the figure legend, was estimated by the Model Finder function of IQ-TREE [31]. The tree was evaluated by a bootstrap analysis with 1000 repetitions [32].

## 3. Results

### 3.1. Detection of Mycoviruses in Turfgrass Pathogenic Colletotrichum Strains

dsRNAs were extracted from the mycelium of nine *Colletotrichum* strains isolated from the diseased turfgrass samples (Table 1) and electrophoresed on an agarose gel. Clear and distinct dsRNA bands were found in two *Collectotrichum* strains, MBCT-264 and MBCT-288 (Figure 1). The sizes of the dsRNA bands ranged from <1.5 to <8.0 kb. The FLDS of MBCT-264 dsRNA provided 1,536,110 reads, which were assembled into eight putative full-length mycovirus sequences (Appendix A). The conserved 5′ terminal ends (CGTTT) of the S264_5_full, S264_1_full and S264_2_full contigs from the strain MBCT-264 indicated that these contigs were three separate segments of a single virus (Appendix A). These segments (contigs) were 1728 bp, 1388 bp and 1160 bp, respectively. BLASTx analysis revealed that S264_5_full and S264_1_full had the best matches with the RdRp and the hypothetical protein of Colletotrichum gloeosporioides partiti-virus 1 (CgPV1), respectively (Table 2 and Appendix A). However, there was no significant similarity match found in the third segment, S264_2_full. We tentatively designated this tri-segmented virus as Colletotrichum associated partitivirus 1 (CaPV1). One-step RT-PCR using primer pairs S264_5_full_FP/RP, S264_1_full_FP/RP and S264_2_full_FP/RP, with a dsRNA from MBCT-264 as a template successfully amplified expected sizes of PCR products of dsRNA-1, dsRNA-2 and dsRNA-3 segments of the virus (Appendix A, Appendix A). Direct sequencing of the amplified product confirmed that the sequences of these RT-PCR products were the same as the FLDS sequences.

A second potential partitivirus with four genome segments, represented by four individual contigs: S264_3_full, S264_4_full, S264_7_full and S264_6_full, was discovered using FLDS in the dsRNA of MBCT-264 (Appendix A). They were 1958 bp, 1872 bp, 1803 bp and 1755 bp long, respectively, and had conserved 5′ ends (AGAATTTCTT) and 3′ ends (AAAAAATAAA) (Appendix A). Two out of these four contigs, S264_3_full and S264_4_full, revealed considerable homology with RdRp and CP of Plasmopara viticola lesion associated partitivirus 9 (PvLaPV9) (Table 2). The other two contigs, S264_7_full and S264_6_full, did not show any similarity to other partitivirus, virus-related, or other sequences. We designated this second partitivirus as Colletotrichum associated partitivirus 2 (CaPV2). Specific one-step RT-PCR and the subsequent Sanger sequencing confirmed the four segments of newly identified CaPV2 in dsRNA extracts from the original isolate and five sub-isolates of MBCT-264 (Appendix A), which supported the results of FLDS. 

Contig S264_8_full, whose length was 7233 nt, was discovered from the MBCT-264 and had the greatest match to the RdRp gene of a minus-strand (−) ssRNA virus, Grapevine associated cogu-like virus 2 (GaCLV) or Grapevine laula-virus 2 (GLV2), which is related to the virus that has a minus-strand (−) ssRNA genome (Appendix A). The tentative name Colletotrichum associated negative-stranded RNA virus 1 (CaNSRV1) was given to this potential (−) ssRNA virus. We confirmed the presence of this virus in dsRNAs of the original and five sub-isolates of MBCT-264 by one-step RT-PCR using specific primers S264_8_full_FP/RP, which amplified specific bands of 797 bp and subsequent direct sequencing (Appendix A). Collectively, FLDS identified three mycoviruses (CaPV1, CaPV2 and CaNSRV1) co-infecting *Colletotrichum* sp. MBCT-264. 

From the dsRNA of MBCT-288, a total of 1,445,468 reads were obtained, which resulted in the identification of two potential mycovirus contigs, S288_1_full (2947 nt) and S288_2_full (7203 nt) (Appendix A). BLASTx search revealed that contig S288_2_full had the best match with the RdRp encoded by Fusarium poae negative-stranded virus 1 and other (−) ssRNA viruses, while contig S288_1_full had the highest sequence similarity to the hypothetical protein of Claviceps aff. purpurea (Table 2). We discovered significant similarities in the 5′- and 3′-terminal non-coding sequences between two contigs S288_2_full and S288_1_full, indicating that they are two segments of the putative (−) ssRNA virus, which we tentatively referred to as Colletotrichum associated negative-stranded RNA virus 2 (CaNSRV2). We successfully detected two segments of CaNSRV2 in the original and sub-isolates of MBCT-288 by one-step RT-PCR using specific primers and subsequent direct sequencing (Appendix A).

### 3.2. Genome Characterization and Phylogenetic Analysis of Colletotrichum Associated Partiti-Virus 1 (CaPV1)

Partiti-viruses typically have two dsRNA segments (1.3 to 2.5 kb in length) with or without the poly (A) tail, one of which encodes RdRp and the other CP [33], although the presence of one or more additional dsRNA segments has been documented in several partitiviruses. CaPV1 had three genome segments, lacked the poly(A) tail and the sizes of dsRNA-1 (1728 bp) and dsRNA-2 (1388 bp) were consistent with the typical genome sizes of RdRp- and CP-coding segments of other partitiviruses but not alphapartitiviruses. dsRNA-1 (48.1% G+C content, GenBank accession number OP471414) had one large ORF (ORF1), dsRNA-2 (49.6% G+C content, GenBank accession number OP471413) had two ORFs (ORF2 and ORF3) and dsRNA-3 (49.6% G+C content, GenBank accession number OP471412) had one ORF (ORF4) (Figure 2A). The UTR sequences at the 5′ and 3′ ends of dsRNA-1, -2 and -3 had considerable pairwise similarity between three segments (Figure 2B). The sequence motif 5′-CGTTTT/A-3′ was identified in the 5′-UTR of all three dsRNA segments, indicating a common viral origin for these segments. CaPV1 ORF1 encoded a 524-aa protein with a calculated molecular mass of 60.3 kDa (Figure 2A), which contained the RdRp conserved domain (pfam00680, Interval: 40–474, E-value: 2.10 × 10^−34^) (Figure 2A). A BLASTp search indicated that it had the highest aa identity to RdRp of CgPV1 (GenBank accession number QED88095; E value = 0.0 and 88.95% identity) and PvLaPV4 (GenBank accession number QHD64807.1; E value = 0.0 and 72.78% identity), which are members of the family *Partitiviridae* (Appendix A). Multiple alignments of RdRp amino acid sequences from CaPV1 and related viruses in the *Partitiviridae* family revealed conservation of six distinct conserved motifs (III–VIII) of RdRp (Figure 2C) [34,35,36]. CaPV1 ORF2 was anticipated to encode a 375-aa protein with a molecular mass of 41.2 kDa (Figure 2A). A BLASTp search demonstrated that the ORF2 product exhibited a 21.9 to 73.4% similarity to hypothetical proteins or CP encoded by some partitiviruses. Among these, the best identity match (73.4%) was found with the hypothetical protein of CgPV1 with 100% sequence coverage (Appendix A). BLASTp hits of CaPV1 ORF2 also included CPs of EnaPV7 (query cover 98.0% and 59.9% identity) and AnPV1 (query cover 97.0% and 27.7% identity), indicating that the ORF2 product of CaPV1 may function as a CP. In addition, a multiple sequence alignment of these putative CPs encoded by CaPV1 and closely related viruses in the family *Partitiviridae* also showed several conserved regions throughout the whole protein (Figure 2D). CaPV1 ORF3 partly overlapped with ORF2 and was found on the negative strand of the virus (Figure 2A). It encoded a 140-aa protein with an estimated molecular mass of 15.1 kDa (Figure 2A). The conserved domain searches found no conserved domains in CaPV1 ORF3 and BLASTp found no similarity to any of the proteins in the nr database of NCBI. The shortest RNA segment, dsRNA-3, had ORF4, which was made up of 228 aa. A BLASTp search of the dsRNA-3 sequence revealed no significant similarities to any nucleotide sequences in the GenBank database. However, when we aligned this ORF4 product of CaPV1 with the hypothetical proteins encoded by dsRNA-3 of CgPV1 (QED88098) and PvLaPV4 (QHD64811), several conserved regions appeared, where the most notable was the conserved motif “FQPGRFSPVVEIV” at their 5′ ends (Figure 2E).

Phylogenetic analysis was performed based on the alignment of the deduced aa sequences of the RdRps of CaPV1 and those of other selected partitiviruses (Figure 3A). This tree identified seven clusters that corresponded to the five partitivirus genera recognized by ICTV: *Alphapartitivirus*, *Betapartitivirus*, *Gammapartitivirus*, *Deltapartitivirus* and *Cryspovirus* [37], as well as two additional partitivirus genera that were proposed: Epsilonpartitivirus and Zetapartitivirus [16,38]. Zetapartitivirus’s cluster, which contained CaPV1, split into two separate subclusters (I and II). CaPV1 belongs to the subcluster II of the Zetapartitivirus, where CgPV1 (QED88095) was the most closely related to CaPV1 with an aa identity of 73.3%, along with six other partitiviruses with aa identities ranging from 64.6 to 88.9%. The four partitiviruses in Zetapartitivirus subcluster I, however, had 49.8 to 52.8% aa identities to CaPV1 (Appendix A). The neighboring cluster of the Zetapartiti-virus was the Gammapartitivirus cluster and the partitiviruses in this cluster shared less than 35% aa identities with CaPV1. The aa sequences of the CPs of CaPV1 and other selected partitiviruses were used to create a second phylogenetic tree (Figure 3B). The resultant tree showed a phylogenetic grouping that was essentially similar to the phylogenetic tree based on RdRps, which assigned CaPV1 to the Zetapartitvirus cluster, having between 27.2 and 73.4% aa identity with CPs of other members of this cluster (Appendix A). These findings, along with the current partitivirus species demarcation criteria (cut-off values of 90% and 80% aa identities for RdRp and CP, respectively) [37], led us to propose that CaPV1 is a novel species of the proposed genus “Zetapartitivirus” in the family *Partitiviridae* [37].

### 3.3. Genome Characterization and Phylogenetic Analysis of Colletotrichum Associated Partiti-Virus 2 (CaPV2)

CaPV2 made up of four genomic segments: dsRNA-1 (1958 bp), dsRNA-2 (1872 bp), dsRNA-3 (1803 bp) and dsRNA-4 (1755 bp) (Appendix A, Figure 4A). The GC content of dsRNA-1 (GenBank accession number OP471411), dsRNA-2 (GenBank accession number OP471410), dsRNA-3 (GenBank accession number OP471409) and dsRNA-4 (GenBank accession number OP471408) full-length sequences was 44.1, 47.7, 44.9 and 42.2%, respectively. The segment size of dsRNA-1 (1958 bp) of CaPV2, which encoded RdRp, was within the range of RdRp-encoding segments of other alphapartitiviruses (1866–2027 bp), while the genome size of CaPV2 dsRNA-2 that encoded CP (1872 bp) was relatively larger than those of the previously reported CP-encoding segments (1708–1866 bp) of the same genus. The nucleotide sequences at the 5′-termini of dsRNA-1 and dsRNA-2 shared a high sequence identity (86%) and contained a conserved sequence “GAAUUUC/GAATTTC” (Figure 4B), in which the G adjacent the 5′-termini was followed by an A, U(T), or C but not by another G for the succeeding five or six nucleotide positions, which are distinctive features of partitiviruses [42]. 5′-UTRs of both CaPV2 genome segments also showed a significantly similar secondary structure, which could play a role in segment recognition of the virus (Figure 4C) [43]. Non-A residues caused interruption in A-rich regions in the 3′-termini of dsRNA-1 and dsRNA-2, which was shown in other alphapartitiviruses (Figure 4B) [5].

CaPV2 dsRNA-1 had a large ORF (ORF1, 68–1894 nt) consisting of 608 aa with a predicted molecular mass of 72.1 kDa. The protein encoded by ORF1 contained a conserved RdRp domain cl02808 (pfam00680, E-value, 7.91 × 10^−13^) from aa positions 210 to 530. BLASTp of the deduced amino acid sequence of RdRp of CaPV2 showed a high similarity to the sequences of RdRp of PvLaPV9 (identity-90.6%, coverage-100%, QHD64790.1) [44] and Fusarium solani partitivirus 2 (FsPV2) (identity-87.6%, coverage-99%, BAQ36631.1) [45] (Appendix A). Multiple sequence alignments of RdRp of CaPV2 and RdRps of selected alphapartitiviruses revealed that CaPV2 RdRp contained three conserved motifs (motifs A, B and C) located in the catalytic palm subdomain that are well conserved among other closely related partitiviruses [38,43,46]. CaPV2 dsRNA-2 had a single ORF (ORF2, 76–1701 nt), which encoded a protein (541 aa) with a predicted mass of 60.4 kDa. In the BLASTp search, ORF2 showed similarity to the CP sequences of PvLaPV9 (QHD64805.1), PvLaPV2 (QHD64794.1), Lichen partiti-like RNA virus 1 (LplRV1, BCD56382.1) and other recognized alphapartitiviruses.

In addition, CaPV2 had two extra genome segments called dsRNA-3 (1803 nt) and dsRNA-4 (1755 nt), which were uncommon but have been observed in some partitiviruses (Figure 4A) [42]. CaPV2 dsRNA-3 had two overlapping ORFs (ORF3 and 4) (Figure 4A). Domain searches of ORF3 did not yield any conserved domains related to any previously reported superfamily or any significant matches to previously reported protein sequences. ORF4 consisted of 111 aa (12.0 kDa), with which we found no significant matches to the published aa sequences in the NCBI database. dsRNA-4 had one ORF (ORF5) and the molecular weight of the encoded protein was 58.2 kDa. The BLASTp search for the ORF5-encoded protein did not find any significant similarity to available sequences in the NCBI database, as well as no conserved domain.

Phylogenetic analysis of aa sequences of RdRp of CaPV2 with related members of partiti-viruses revealed that it belongs to the clade encompassing the genus *Alphapartiti-virus* with strong bootstrap support (Figure 3A). The closest phylogenetic neighbors of CaPV2 were PvLaPV9 and FsPV2. The notion that CaPV2 belongs to the genus *Alpha-partitiviruses* supported by the phylogenetic analysis of the CP aa sequences (Figure 3B). Considering that the species demarcation criterion for a new species of partitivirus is 90% aa and 80% sequence identity in RdRp and CP, respectively and that identities between both proteins of CaPV2 and those of other partitiviruses are lower than the criteria, we propose that CaPV2 is a novel species of the genus *Alphapartitivirus* in the family *Partitiviridae*.

### 3.4. Genome Characterization of Colletotrichum Associated Negative-Stranded RNA Virus 1 (CaNSRV1) and Phylogenetic Relationship with Related Viruses

A single genome segment of CaNSRV1 (7233 nt, GenBank accession number OP471407) was found from the *Colletotrichum* strain MBCT-264, which lacked a poly (A) tail region and had an ORF (ORF1) in its complementary (vc) strand. 5′- and 3′-ends of the CaNSRV1 genome were significantly similar to their respective ends in Grapevine associated cogu-like virus 3 (GaCLV3) or Grapevine laula-virus 3 (GLV3), Grapevine associated cogu-like virus 2 (GaCLV2) or Grapevine laula-virus 2 (GLV2) and Laurel Lake virus (LLV) of the *Phenuiviridae* family. ORF1 product consisted of 2382 aa with a predicted molecular weight of 274.0 kDa (Figure 5A). The NCBI domain search of the ORF1-encoded protein of CaNSRV1 identified the RdRp conserved domain of members of the order *Bunyavirales* (pfam04196, Bunya-RdRp; E-value: 2.10 × 10^−35^; Interval: 727–1387 aa) (Figure 5A). BLASTp analysis revealed that CaNSRV1 RdRp is most similar to the RdRp of GaCLV2 and GaCLV3, with pairwise identities of 69.7%, a value below the threshold of a species demarcation criteria (90%) of most genera in the order *Bunyavirales* (Appendix A). Furthermore, aa alignments of RdRp of CaNSRV1 and related viruses verified the presence of the six conserved motifs of Bunyavirales RdRps, including pre-motif A and motifs A to E. CaNSRV1 possessed the motifs A (DATKWC), B (QGILHYTSS), C (SDD) and D (KS) (Figure 5B). The motif E had tetrapeptide E(F/Y)xS, which was seen in the polymerases of segmented negative-sense RNA viruses [47,48]. The basic residues K, R and R/K in pre-motif A, as well as a glutamic acid (E) downstream of pre-motif A, were also found to be conserved [34,49]. The N-terminal region, located upstream of pre-motif A of the RdRp of CaNSRV1, contained the endonuclease conserved motif involved in cap-snatching (H_218_D_228_PD_245–246_ExT_257–260_K_276_), a strategy used by many negative-stranded viruses to translate viral proteins by using capped terminal ends from host mRNAs (Figure 5C) [50]. These findings indicate that ORF1 product of CaNSRV1 represents RdRp of negative-stranded ssRNA viruses, especially bunyaviruses. A phylogenetic tree revealed that the RdRp of CaNSRV1 formed a monophyletic group with those of LLV, GaCLV2, GaCLV3 and GaCLV4, which belong to the genus *Laulavirus*. Furthermore, the clade containing CaNSRV1 and these laulaviruses was a part of a superclade that also included a cogu-virus clade (Figure 5D), which contained Citrus concave gum-associated virus (CCGaV), citrus virus A (CVA) and watermelon crinkle leaf-associated virus-1 and -2 (WMLaV1 and 2), members of the Coguvirus group that infect plants. Therefore, we conclude that CaNSRV1 is a novel member of the genus *Laulavirus* in the family *Phenuiviridae*.

### 3.5. Genomic Characterization of Colletotrichum Associated Negative-Stranded RNA Virus 2 (CaNSRV2) and Phylogenetic Relationship with Related Viruses

CaNSRV2 genome consisted of two RNA segments: RNA-1 (7203 nt, GenBank accession number OP471406) and RNA-2 (2947 nt, GenBank accession number OP471405) and no poly (A) tail regions were detected in both segments (Figure 6A). We found 68% identity between the 5′ terminal ends of RNA-1 and RNA-2 and 70% identity between the 3′ terminal ends of RNA-1 and RNA-2 (Figure 6B). RNA-1 had three ORFs (ORF1, 2 and 3), which were >0.3 kb. ORF1 (174 aa) and ORF2 (129 aa) were positioned on the positive strand and overlapped with ORF3 (2346 aa) placed on the complementary strand. ORF3 encoded a putative RdRp with an estimated molecular weight of 269.9 kDa, which contained the RdRp conserved domain of Mononegavirales (pfam00946; predicted at 407-808 aa, E-value of 4.62 × 10^−16^) (Figure 6A). The putative RdRp of CaNSRV2 shared 65.6% identity with the RdRp of Fusarium poae negative-stranded virus 1 (FpNSRV1, YP 009272911.1) (Appendix A) [51]. Multiple alignment of the RdRp sequences of CaNSRV2 and related viruses revealed five conserved motifs—pre-motif A, A, B, C and D. Pre-motif A (K_575_EREQKYEARLF_586_) of CaNSRV2 was 100% identical to that of FpNSRV1 and 91% identical to Cladosporium cladosporioides negative-stranded RNA virus 1 (CcNSRV1), with perfect conservation of the lysine (K), arginine (R) and glutamic acid (E) residues. The motif-A (S_643_LLLDIEGHNQSMQ_656_) of CaNSRV2 exhibited similarities with the motif A of ophiviruses, which included conserved leucine (L), aspartic acid (D), glycine (G), histidine (H), asparagine (N) and serine (S) residues and was 100% identical to FpNSRV1 and 92.8% identical to CaNSRV2. The glycine residues were conserved in motif B of CaNSRV2 and other related viruses. Motif C, found in the RdRp of CaNSRV2 and comparable viruses, contained the YSDD signature, which was also found in *Orthomyxoviridae* members and corresponded to the GDNQ signature found in the majority of negative-stranded RNA viruses, which acts as an RdRp active site (Figure 6C). In motif D, a glycine residue was conserved in CaNSRV2 and closely related myoophioviruses, including FpNSRV1 and CcNSRV1. With the help of the online software cNLS Mapper, we discovered four bipartite nuclear localization signals (NLSs) in the predicted RdRp protein of CaNSRV2 [52] (Figure 6D). NLSs were also found to be conserved in other closely related mycoophioviruses, as well as those of ophioviruses infecting plants. Phylogenetic analysis of aa sequences of RdRps of CaNSRV2 with those of viral species from the orders *Serpentovirales*, *Mononegavirales* and *Jingchuvirales* [44] has revealed that CaNSRV2 belongs to the same lineage as ophioviruses. However, CaNSRV2, together with ophio-like viruses infecting plant pathogenic fungi, clustered as a distinct clade within the ophio-virus superclade (Figure 6E), which was recently proposed for a new family called the mycoaspiviridae [33].

Another segment of the identified genome of CaNSRV2 was predicted to possess four ORFs (>0.3 kb; ORF 5, 6, 7 and 8) (Figure 6A). Among the ORFs in the second genome of CaNSRV2, only ORF7 exhibited considerable similarity to the hypothetical protein sequences encoded by Claviceps purpurea in the NCBI nr database (Appendix A).

## 4. Discussion

The goal of this work was to discover mycoviruses in turfgrass-pathogenic *Colletotrichum* spp. in Japan in order to develop bio-control agents in the future. Although several viruses have been identified in *Colletotrichum* spp. isolated from other host plants, no mycoviruses have been reported to date in *Colletotrichum* spp. pathogenic to turfgrass. Using screening employing dsRNA electrophoresis combined with FLDS, we successfully identified four new RNA viruses from *Colletotrichum* spp. infecting turfgrass in Japan (Appendix A). FLDS has been used with high sensitivity to detect multiple viruses infecting the same host at the same time, as well as to distinguish segmented genomes of viruses [21]. In this study, FLDS-generated reads were de novo assembled into eight full-length viral RNA genomes from *Collectotrichum* sp. MBCT-264 and two full-length viral RNA genomes from *Colletotrichum* sp. MBCT-288 (Appendix A). Eight viral sequences from MBCT-264 were considered to be two novel dsRNA partitiviruses: CaPV1 that had three segments and CaPV2 that had four segments, as well as a negative single-stranded ssRNA virus that had a single genome segment: CaNSRV1. Two viral sequences from MBCT-288 were classified as two segments of a novel negative single-stranded ssRNA virus, CaNSRV2. To our knowledge, this is the first report of mycoviruses detected in turfgrass pathogenic *Colletotrichum* spp.

We proposed that CaPV1 is a new member of the proposed genus “Zetapartitivirus” in the *Partitiviridae* family, based on the phylogenetic analysis (Figure 3A) [35]. Most partitiviruses have two dsRNA genome segments, which encode RdRp and CP, while some viruses in the *Partitiviridae* family have additional dsRNA segments [42]. Actually, CaPV1 was predicted to have an additional segment, dsRNA-3, in addition to the RdRp-encoding dsRNA-1 and CP-encoding dsRNA-2. In the proposed genus “Zetapertitivirus,” Aspergillus flavus partiti-virus 1 (AfPV1) and Botryosphaeria dothidea virus 1 (BdV1) also have three genomic segments, of which the function of one segment remains unknown, but the genomes of Colletotrichum acutatum RNA virus 1 (CaRV1), Ustilaginoidea virens partiti-virus 2 (UvPV2) and UvPV3 each have only two segments, encoding RdRp and CP [35]. The absence of an additional segment in some of the zetapartitiviruses may be due to technical problems; the additional segments would be found if highly sensitive and specific FLDS were employed.

Although members of the *Partitiviridae* family generally have a latent effect on their hosts [17,42,53], a few partitiviruses are reported to cause abnormal phenotypes or a reduction in virulence in their hosts. For example, Aspergillus fumigatus partitivirus 1 (AfuPV1) of the genus *Gammapartitivirus* causes abnormal colony phenotypes, slow growth and light pigmentation of its host. The presence of Flammulina velutipes browning virus (FvBV) of the genus *Alphapartitivirus* has been linked to the color of fruiting bodies of the host fungus [54,55,56]. Notably, AfPV1, which belongs to the same proposed genus Zetapartitivirus as CaPV1 (Appendix A), causes debilitating symptoms such as abnormal colonial morphology, slow growth, poor sporulation and short spore chains, despite having no significant effect on host virulence [55,57]. It remains to be addressed in a future study whether CaPV1 causes some effects on its host.

This study discovered the co-infection of another partitivirus, CaPV2, with CaPV1 described above, in *Colletotrichum* sp. MBCT-264. We proposed that CaPV2 is a new species of the genus *Alphapartitivirus* based on the sequence identities of RdRp and CP genes with other members of the genus. Alphapartitiviruses typically have two dsRNA segments of similar size, but CaPV2 was discovered to have four genome segments, with only the RdRp- and CP-encoding segments showing considerable sequence similarity to other partiti-viruses (Appendix A) [42]. A third genomic segment of the alphapartiti-virus Rosellinia necatrix partitivirus 2 (RnPV2) was reported to be a truncated variant of the RdRp that may act as defective-interfering RNA [58], but dsRNA-3 and -4 of CaPV2 were not defective segments of dsRNA-1 or -2 and it remains unknown what function dsRNA-3 and -4 possessed. It was found that the 5′ terminal sequence of alphapartitivirus RnPV2 was conserved and predicted to form a stem–loop structure. The 5′-terminal sequences of the CaPV2 segments were likewise significantly conserved and formed stable RNA stem–loop structures [59]. Therefore, the conserved sequence and stem–loop structure of CaPV2 are likely to play a role in replication and assembly of the virus (Figure 5C). Also, there are adenine-rich regions in the 3′ terminal ends, referred to as ‘interrupted’ poly A tails, of CaPV2 dsRNA-1 and dsRNA-2, which have been found in other alphapartitiviruses and are responsible for virus replication [59,60].

We discovered CaNSRV1, a virus related to laulaviruses, a newly characterized group of plant viruses in the family *Phenuiviridae* [44,61,62], which is a well-studied family of negative-stranded viruses in the *Bunyavirales* order. Viruses belonging to this order have been found in vertebrates, invertebrates and plants, as well as in fungi, from which they have recently been discovered [63,64,65]. Generally, negative-stranded viruses have rarely been discovered in fungi, with only a few of them having been characterized in detail [66]. In fungal hosts, even when RdRp-encoding bunya-like sequences have been reported, the associated segments such as those encoding nucleocapsid protein (Nc) or others are often missing [66]. Consistent with this, no protein-coding contigs other than those encoding bunya-like RdRp were discovered from the isolate MBCT-264. In contrast, in the previously reported plant-infecting laula-viruses, segments encoding Nc and MP were also found [44,61]. Coguviruses infecting plants and Botrytis cinerea bocivirus 1 (BcBV1) appear to have a three-segmented genome, with RNA1, RNA2 and RNA3 encoding the putative RdRp, MP and Nc, respectively [60,67]. As a result, CaNSRV1 is distinct from other laulaviruses and cogu-viruses in aspects of genome structure other than the RdRp-encoding segment. This finding led us to wonder how viruses with such disparate genomic structures could have evolved from a common ancestor. One possible explanation is given by Koonin and colleagues, who suggested that a common ancestor of groups of RNA viruses may have only the RdRp-encoding part and MP and Nc were acquired later independently [68]. Therefore, we would claim that CaNSRV1 is a mono-segmented virus, although we cannot exclude the possibility that we missed some of its genomic segments in this study.

We also discovered CaNSRV2 from *Colletotrichum* sp. MBCT-288, another negative-stranded RNA virus with an ophio-virus-like RdRp. Ophioviruses are plant viruses that belong to the *Aspiviridae* family in the order *Serpentovirales*, which have segmented negative-stranded RNA genomes with up to four segments. CaNSRV1 belonged to the recently proposed “Mycoaspiviridae” family clade together with several mycoviruses, including FpNSRV1 and others [33,51,62]. The family “Mycoaspiviridae” has considerable genetic distance from the *Aspiviridae* family clade. Surprisingly, we discovered a second genome segment for CaNSRV1, although the previously reported mycoophioviruses were single-segmented and only encoded RdRp. This is consistent with the hypothesis by Chiapello and colleagues (2020) that mycophio-viruses might have extra genomic segments associated with them, which has yet to be confirmed [44]. CaNSRV2, along with other mycoophiviruses and ophioviruses, may have descended from a common ancestor.

As mentioned above, neither CaNSRV1 nor CaNSRV2 have MP or Nc, although MP may not be required for their replication in their fungal host. Given that nucleocapsid formation is a requirement for (−) ssRNA virus replication, it will be interesting to investigate how viruses such as CaNSRV1 or CaNSRV2 replicate in their fungal hosts.

Furthermore, we report the coexistence of two partitiviruses and a negative-stranded ssRNA virus in *Colletotrichum* sp. MBCT-264. Multiple viral infections are common in fungi [7,69] and a single fungal strain can be infected simultaneously by several unrelated viruses [51,70]. In an Asian *R. necatrix* isolate, Chun and colleagues reported the presence of a fusagravirus, Rosellinia necatrix fusagra-virus 4 and its co-infection with a partiti-virus, Rosellinia necatrix partiti-virus 26 (RnPV26) [71]. The Japanese *R. necatrix* strain W442 was reported to co-infect with the partitiviruses RnPV18 (of the genus *Betapartiti-virus*) and RnPV19 (of the genus *Alphapartiti-virus*) [71]. Viruses co-infecting the same host may result in synergistic, mutualistic, or antagonistic virus–virus interactions. One virus can increase the concentration of another co-infecting virus when there is a mixed infection. Additionally, co-infecting viruses can also reduce vegetative incompatibility, increasing the horizontal spread of viruses via hyphal anastomosis [62,72]. Whether the co-infecting mycoviruses in *Colletotrichum* sp. MBCT-264 interact with one another and have any effect on the host is unknown. Our findings showed that a single *Colletotrichum* strain could be infected with multiple mycoviruses. We need to observe mycovirus diversity in other *Colletotrichum* spp. that infect grasses and other gramineous plants since it is interesting to see distinct patterns of virus infection in *Colletotrichum* spp. MBCT-264 and *Collectotrichum* sp. MBCT-288. Further research into the biological effects of the mycoviruses discovered in this study is required, because we did not perform horizontal transmission experiments in this study. Identifying hypovirulent mycoviruses and determining their potential use in biocontrol programs is appealing because reducing the impact of *Colletotrichum* species in turfgrasses is important for maintaining the green beauty of golf fields. It is also worth looking into the role of mycoviruses on turfgrasses–*Colletotrichum* interactions, as this could reveal new information about the pathogenicity mechanism of *Colletotrichum* spp.

## Figures and Tables

**Figure 1 viruses-14-02572-f001:**
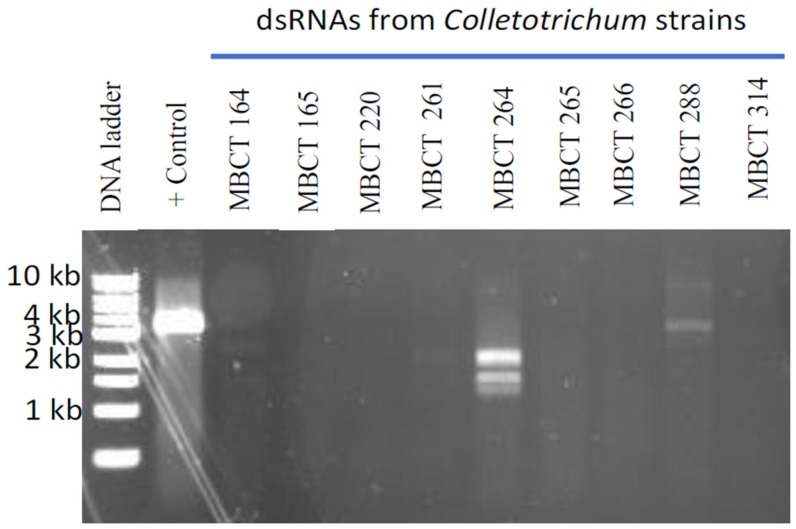
Agarose gel electrophoresis of dsRNA isolated from nine Colletotrichum isolates. As a positive control, we used dsRNA from Magnaporthe oryzae chryso-virus 1 (MoCV1) A infected Magnaporthe oryzae isolate. Clear and distinct dsRNA bands were found in two Collectotrichum isolates, MBCT-264 and MBCT-288.

**Figure 2 viruses-14-02572-f002:**
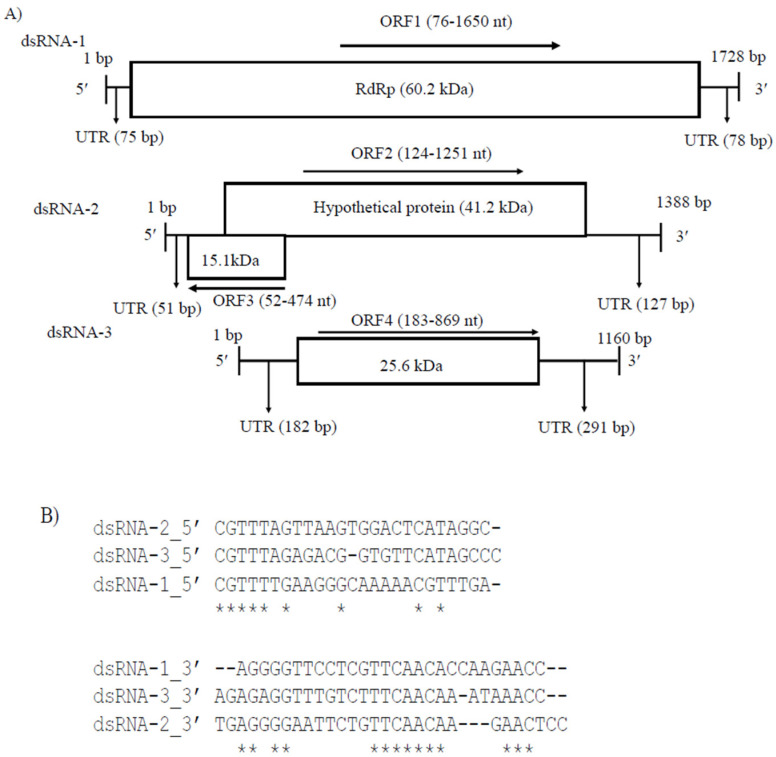
(**A**) Schematic representation of the CaPV1 genome structure. ORF1 encodes an RdRp and ORF2 encodes a hypothetical protein (comparable to coat protein, CP). (**B**) Comparison of the 5′- and 3′-terminal sequences of dsRNA-1, dsRNA-2 and dsRNA-3. (**C**) Multiple alignment of the sequences of the conserved motifs in the RdRps of CaPV1 and other partitiviruses. The six conserved motifs are indicated by the colored boxes. (**D**) Several conserved regions were discovered in a multiple sequence alignment of hypothetical proteins encoded by CaPV1 and closely related viruses from the Partitiviridae family. (**E**) Alignments of ORF4 product of CaPV1 with the hypothetical proteins encoded by dsRNA-3 of CgPV1 (QED88098) and PvLaPV4 (QHD64811).

**Figure 3 viruses-14-02572-f003:**
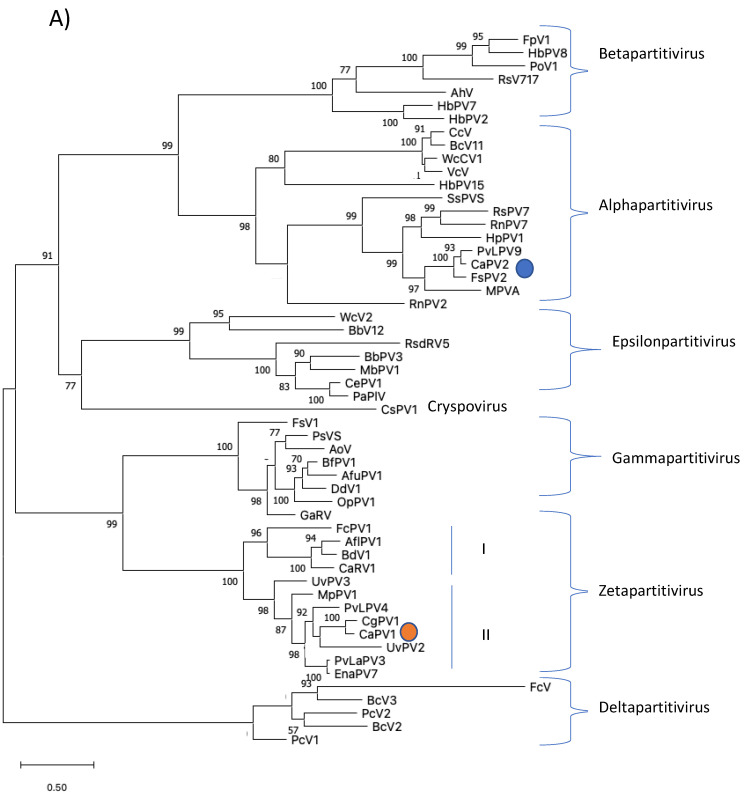
Phylogenetic analyses of RdRps (**A**) and CPs (**B**) of CaPV1, CaPV2 and related partiti-viruses using the function “build” of ETE3 3.1.2 [39] as implemented on the GenomeNet (https://www.genome.jp/tools/ete/, accessed on 2 September 2022). Alignment was performed with MAFFT v6.861b with the default options [40]. ML tree was inferred using IQ-TREE 1.5.5 ran with ModelFinder and tree reconstruction [41]. Best-fit models, according to BIC, were VT+F+R5 and VT+F+I+G4 for RdRp and CP phylogenetic trees, respectively. Tree branches were tested by SH-like aLRT with 1000 replicates. Different members of the partiti-virus genera, including Alpha-, Beta-, Delta-, Gamma- and the proposed Epsilonpartitivirus were included in the analyses. Viruses found in this study were marked with a circle. Bootstrap values are indicated at the branches. The scale bar (lower left) represents the genetic distance of 0.50 for the phylogenetic tree of partitiviral RdRps (**A**) or CPs (**B**). Full names and GenBank accession numbers of the viruses listed in Appendix A.

**Figure 4 viruses-14-02572-f004:**
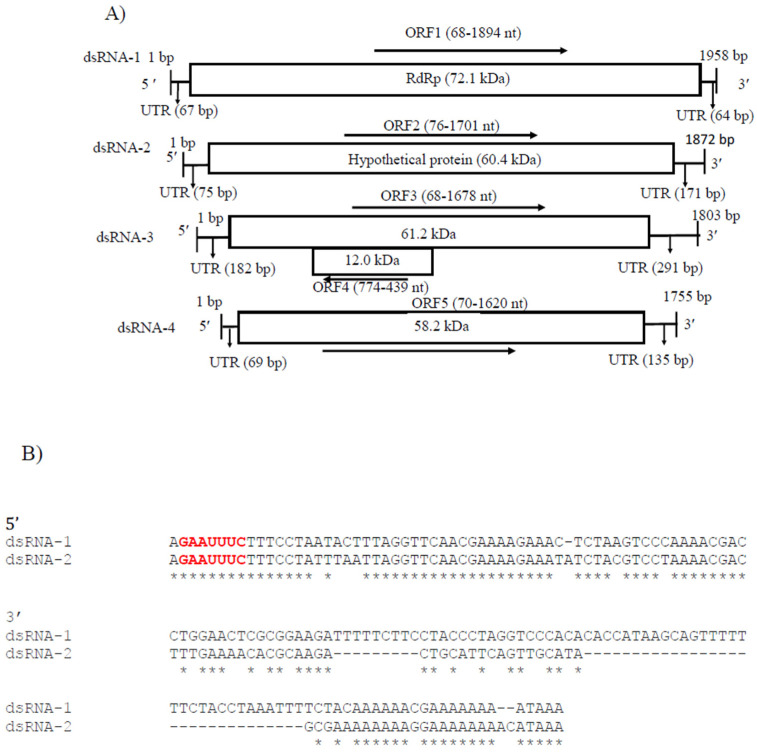
(**A**) Schematic representation of the CaPV2 genome organization. All the genome segments contain a single ORF (rectangular box) except dsRNA-3. dsRNA-1 encoded an RNA-dependent RNA polymerase (RdRp) and dsRNA-2 encoded a capsid protein (CP). The untranslated regions (UTRs) at both termini of dsRNA segments are also shown. (**B**) Sequence alignment of the respective 5′ UTRs between dsRNA-1 and dsRNA-2 of CaPV2. (**C**) Predicted secondary structure of the 5′ UTR sequences of CaPV2 dsRNA-1 (left) and dsRNA-2 (right). (**D**) Multiple sequence alignment analysis of RdRp from different members of the genus *Alphapartitivirus* revealed that CaPV2 RdRp contains all three conserved motifs (motifs A–C) found in the catalytic palm subdomain and these motifs are well conserved among dsRNA viruses. The conserved motifs are indicated by the colored boxes.

**Figure 5 viruses-14-02572-f005:**
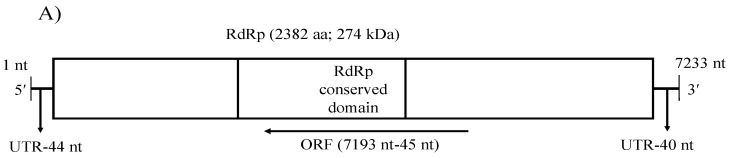
(**A**) Schematic representation of the CaNSRV1 genome organization. (**B**) The presence of the six conserved motifs, which included pre-motif A and motifs A–E, was verified by aa alignments in RdRps of CaNSRV1 and other related (-ss) RNA viruses, which match to highly conserved sections of the order Bunyavirales RdRps [34,47,48,49]. CaNSRV1 comprises the motifs A (DATKWC), B (QGILHYTSS), C (SDD), D (KS) and E (E(F/Y)xS). E is a tetrapeptide motif found in the RdRp of segmented negative-sense RNA viruses. Furthermore, three basic residues in pre-motif A: K, R and R/K, as well as a glutamic acid. The conserved motifs are indicated by the colored boxes. (E) downstream of pre-motif A, were shown to be conserved in CaNSRV1 and related viruses RdRps. (**C**) N-terminal region of CaNSRV1 had an endonuclease conserved motif that was engaged in cap-snatching, a method adopted by many negative-stranded viruses to translate viral proteins by utilizing capped terminal ends of host mRNAs. The ExT domain conserved in the RdRp of most bunyaviruses is also found in CaNSRV1. (**D**) Alignment and phylogenetic reconstructions of RdRps of CaNSRV-1 and selected viruses were performed using the function “build” of ETE3 3.1.2 [39] as implemented on the GenomeNet (https://www.genome.jp/tools/ete/, accessed on 2 September 2022). Alignment was performed with MAFFT v6.861b with the default options [40]. ML tree was inferred using IQ-TREE 1.5.5 ran with ModelFinder and tree reconstruction [41], where best-fit model according to BIC was LG+F+R5. Tree branches were tested by SH-like aLRT with 1000 replicates. CaNSRV1 was marked with a circle. CaNSRV1 is a member of a superclade that also comprises the coguvirus clade, which is made up of viruses from the coguvirus genus. The scale bar (lower left) represents the genetic distance of 0.50 for the phylogenetic tree of RdRps.

**Figure 6 viruses-14-02572-f006:**
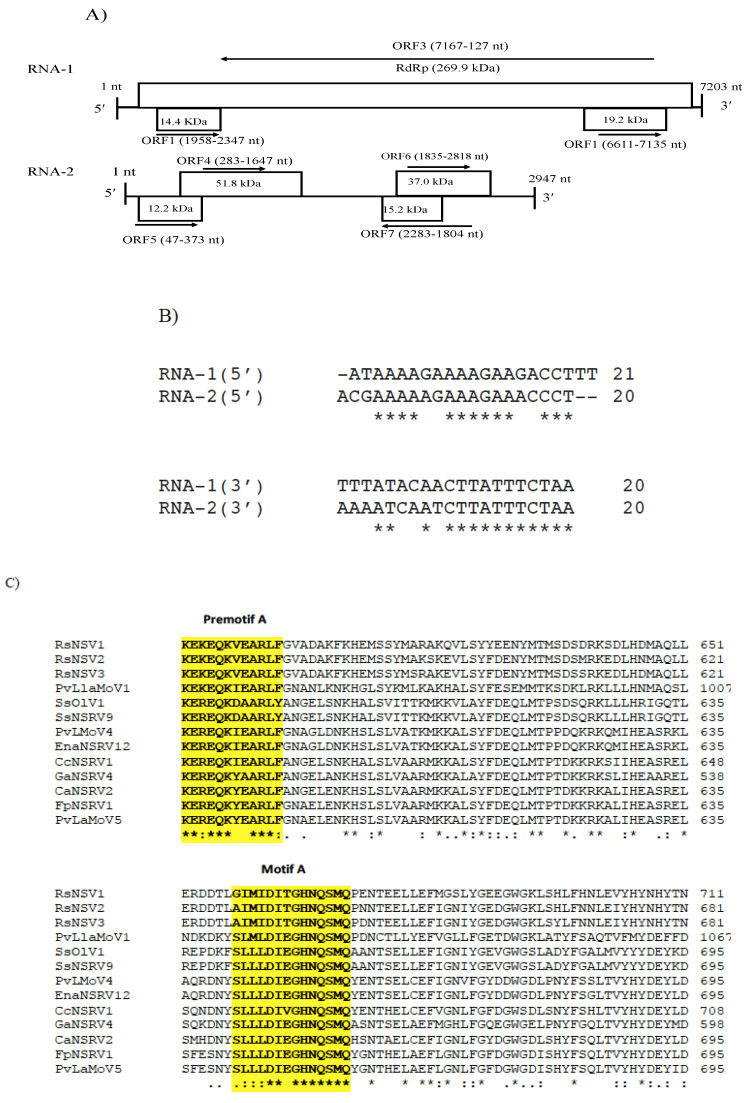
(**A**) Schematic representation of the CaNSRV2 genome organization. (**B**) Sequence alignments of the terminal ends of CaNSRV2 RNA-1 and RNA-2. (**C**) The multiple alignment of the RdRp sequences of CaNSRV2 and the other related viruses, which indicated four conserved motifs: motif A(SLLLDIEGHNQSMQ), motif B (QLGGIEGWLNPLWTL), motif C (YSDD) and motif D (ADGIRADSTLKRL) and pre-motif A (KEREQKYEARLF). (**D**) Bipartite nuclear localization signals (NLSs) predicted in the RdRp protein of CaNSRV2. (**E**) Alignment and phylogenetic reconstructions were performed using the function “build” of ETE3 3.1.2 [39] as implemented on the GenomeNet (https://www.genome.jp/tools/ete/, accessed on 2 September 2022). Alignment was performed with MAFFT v6.861b with the default options [40]. ML tree was inferred using IQ-TREE 1.5.5, run with ModelFinder and tree reconstruction [41]. The best-fit model, according to BIC was LG+F+R5. Tree branches were tested by SH-like aLRT with 1000 replicates. CaNSRV2 was marked with a circle. The scale bar (lower left) represents the genetic distance of 1 for the phylogenetic tree of RdRps.

**Table 1 viruses-14-02572-t001:** *Colletotrichum* isolates used for dsRNA isolation in this study.

Strain	Host	Collection Place	Identified Pathogen
MBCT-164	Bermuda grass (*Cynodon dactylon*)	Ibaraki Prefecture	*Colletotirchum* sp.
MBCT-165	Bermuda grass (*Cynodon dactylon*)	Ibaraki Prefecture	*Colletotirchum* sp.
MBCT-220	Manila grass (*Zoysia matrella*)	Saitama Prefecture	*Colletotirchum* sp.
MBCT-261	Manila grass (*Zoysia matrella*)	Tochigi Prefecture	*Colletotirchum* sp.
MBCT-264	Manila grass (*Zoysia matrella*)	Shizuoka Prefecture	*Colletotirchum* sp.
MBCT-265	Japanese lawngrass (*Zoysia japonica*)	Shizuoka Prefecture	*Colletotirchum* sp.
MBCT-266	Japanese lawngrass (*Zoysia japonica*)	Shizuoka Prefecture	*Colletotirchum* sp.
MBCT-288	Creeping bentgrass (*Agrostis stolonifera*)	Ibaraki Prefecture	*Colletotirchum* sp.
MBCT-314	Creeping bentgrass (*Agrostis stolonifera*)	Ibaraki Prefecture	*Colletotirchum* sp.

**Table 2 viruses-14-02572-t002:** Assembled sequences with best similarity in BLASTx to those of previously described viruses.

SL	Contig	Best Match	% aa Identity	Query Cover (%)	Genome Type/Putative Gene	Family or Genus
FLDS of dsRNA of MBCT-264						
1	S264_5_full	*Colletotrichum gloeosporioides* partitivirus 1, CgPV1	88.95	90	dsRNA/RdRp	Partitiviridae
2	S264_1_full	CgPV1	73.4	73.4	dsRNA/hypothetical protein/CP	Partitiviridae
3	S264_2_full	No significant similarity found				
4	S264_3_full	*Plasmopara viticola* lesion associated Partitivirus 9, PvLaPV9	90.62	93	dsRNA/RdRp	Partitiviridae
5	S264_4_full	PvLaPV9	73.74	86	dsRNA/CP	Partitiviridae
6	S264_7_full	No significant similarity found				
7	S264_6_full	No significant similarity found				
8	S264_8_full	Grapevine associated cogu-like virus 2, GaCLV2	69.78	98	(−) ssRNA/RdRp	*Phenuiviridae*
FLDS of dsRNA of MBCT-288						
9	S288_2_full	Fusarium poae negative-stranded virus 1, FpNSV1	65.69	97	(−) ssRNA/RdRp	unclassified ssRNA negative-strand viruses
10	S288_1_full	Claviceps aff. purpurea	54.58	30	(−) ssRNA/hypothetical protein	unclassified ssRNA negative-strand viruses

## Data Availability

All data obtained in this research is available upon request.

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
