# Peer review of "Discovery, Genomic Sequence Characterization and Phylogenetic Analysis of Novel RNA Viruses in the Turfgrass Pathogenic Colletotrichum spp. in Japan"

_viruses, 2022, doi:10.3390/v14112572_

Round 1
Reviewer 1 Report
The manuscript describes 4 mycoviruses discovered through a dsRNA sequencing approach, and provides analysis on full genome sequences of these mycoviruses. It demonstrates the advantage of the approach in determining the ends of viral genomes that produce large amounts of dsRNA replicative forms. However, to find potential biological control candidates, this reviewer believes that an overall metatranscriptomic survey of the total RNA is going to provide a better resolution in the diversity of mycoviruses that these fungal strains harbor. This reviewer suggests to pool all the RNAs for 1 run of RNA-Seq, and traced back to the individual RNA extract for a given viral contig after the bioinformatic analysis. It will provide an overall view of the mycoviruses on top of the existing manuscript, even though the ends may or may not be complete. It is very cost effective and will shed light on the diverse viruses infecting these selected strains that could be related to hypovirulence, because FLDS simply cannot "comprehensively investigate viromes", leaving out many viruses that don't produce a lot of dsRNA. As a minor point, many grammatic and spelling errors can be found throughout the manuscript, and will need careful revision.
Author Response
Dear Reviewer 1,
We are grateful to the reviewer for helping us improve our manuscript, ID: viruses-1953051. Those revisions are all valuable and very helpful for revising and improving our paper. Please see below our responses to each of the comments.
>1. The manuscript describes 4 mycoviruses discovered through a dsRNA sequencing approach, and provides analysis on full genome sequences of these mycoviruses. It demonstrates the advantage of the approach in determining the ends of viral genomes that produce large amounts of dsRNA replicative forms.
Ans: Successful detection of known and novel RNA viruses from a wide range of organisms or microorganisms using FLDS, an efficient dsRNA targeting metagenomic tool, has already been demonstrated in several independent studies (Urayama et al. 2016, Urayama et al. 2018, Chiba et al. 2021, Hirai et al. 2022) . The recovery of terminal sequences, uniform read coverage, and an incredibly high recovery rate of viral RNA sequences, including dsRNA and ssRNA viruses, are all features of this FLDS method that have not been previously reported in other metagenomic methods such total RNA sequencing targeting RNA viruses (Urayama et al. 2016).Therefore, our primary goal in this study is not to demonstrate the benefit of FLDS in identifying the ends of viral genomes that produce significant amounts of dsRNA replicative forms. Our main goal in this study is to discover, characterize, and phylogenetic analysis of novel RNA viruses that are present in turfgrass pathogenic Collectotrichum spp. in Japan.To address this issue, we believe that our strategy using mycovirus screening by dsRNA electrophoresis and subsequent FLDS is reasonable and sufficient. In order to reduce the emphasis on the efficiency of FLDS, we deleted “that can detect a wide range of RNA viruses” from the first part of the Discussion section (line 730), and we deleted the part shown below from the Discussion section (line 876). “, implying that the FLDS method can be used to comprehensively investigate viromes in phytopathogenic fungi”
References:
Urayama, S.I., Takaki, Y. and Nunoura, T., 2016. FLDS: a comprehensive dsRNA sequencing method for intracellular RNA virus surveillance. Microbes and environments, p.ME15171.
Urayama, S.I., Takaki, Y., Nishi, S., Yoshida‐Takashima, Y., Deguchi, S., Takai, K. and Nunoura, T., 2018. Unveiling the RNA virosphere associated with marine microorganisms. Molecular ecology resources, 18(6), pp.1444-1455.
Chiba, Y., Oiki, S., Yaguchi, T., Urayama, S.I. and Hagiwara, D., 2021. Discovery of divided RdRp sequences and a hitherto unknown genomic complexity in fungal viruses. Virus evolution, 7(1), p.veaa101.
Hirai, J., Urayama, S.I., Takaki, Y., Hirai, M., Nagasaki, K. and Nunoura, T., 2022. RNA Virosphere in a Marine Zooplankton Community in the Subtropical Western North Pacific. Microbes and environments, 37(5), p.ME21066.
>2. However, to find potential biological control candidates, this reviewer believes that an overall metatranscriptomic survey of the total RNA is going to provide a better resolution in the diversity of mycoviruses that these fungal strains harbor. This reviewer suggests to pool all the RNAs for 1 run of RNA-Seq, and traced back to the individual RNA extract for a given viral contig after the bioinformatic analysis. It will provide an overall view of the mycoviruses on top of the existing manuscript, even though the ends may or may not be complete. It is very cost effective and will shed light on the diverse viruses infecting these selected strains that could be related to hypovirulence, because FLDS simply cannot "comprehensively investigate viromes", leaving out many viruses that don't produce a lot of dsRNA.
Ans: We totally agree with this reviewer’s comment that FLDS could not provide the complete RNA virome in the samples. FLDS only has the ability to detect ssRNA viruses at the replicative stage. Meanwhile, as suggested by previous studies, metatranscriptomic survey of the total RNA by RNAseq could not either solve the complete virome. Indeed, the viral read ratio of intracellular RNA viruses in the RNA-seq library is often less than 1% since mRNA and rRNA dominate in the total RNA fraction isolated from biological samples (Matranga et al. 2014), while FLDS enhanced the viral RNA reads by over 300-fold when compared to total RNA-seq (Urayama et al. 2016) . As suggested, combination of FLDS and RNAseq may clarify virome with higher solution, but the focus of this study is to discover and characterize the RNA viruses in the Colletotrichum sp. in turfgrass, not to reveal the comprehensive virome of the organism. Complete identification of fungal RNA virome, as well as DNA virome, is out of the scope of this study. Because pooling all the RNAs for a single RNAseq run will make it more challenging to identify viruses with multiple segments and viruses that are co-infected, it is reasonable and sufficient for us to employ FLDS of dsRNA samples.To clarify these points, we modified and deleted some parts from Discussion. Again, pooling all the RNAs for a single RNA-Seq run may be economical, but it requires extensive bioinformatic analysis, and when a viral contig is traced back to its individual RNA extract, extensive and time-consuming RT-PCR and sanger sequencing are required, which raises the cost of the investigation.
Reference:
Matranga, C.B., K.G. Andersen, S. Winnicki, et al. 2014. Enhanced methods for unbiased deep sequencing of Lassa and Ebola RNA viruses from clinical and biological samples. Genome Biol. 15:519.
Urayama, S.I., Takaki, Y. and Nunoura, T., 2016. FLDS: a comprehensive dsRNA sequencing method for intracellular RNA virus surveillance. Microbes and environments, p.ME15171.
>3.As a minor point, many grammatic and spelling errors can be found throughout the manuscript, and will need careful revision.
Ans: Grammar and spelling errors were thoroughly checked all through the entire manuscript.
Reviewer 2 Report
This study for the first time reports mycoviruses in Colletotrichum spp. infecting turfgrass in Japan. In addition, this study reports four new mycoviruses, including one new Alphapartitivirus and Zetapartitivirus, and two new -ssRNA viruses. This article can be accepted after the authors have agreed to alter the manuscript as suggested.
In the summary, check the proper spacing between words carefully. Italicize Partitiviridae as it indicates a virus family here. Check throughout the manuscript and wherever you mention an ICTV-recognized virus species, genus or family, italicize the name.
Line 23: make it Colletotrichum-associated negative-stranded RNA virus 1 (CaNSRV1)
Line 66: it is chestnut blight
Line 110: ITS-Internal transcribed spacer
In the materials and methods section, the authors said that they collected nine isolates (line 105) but Table1 shows 11 isolates. Why this discrepancy?
Regarding mycovirus sequencing, I am wondering why the authors chose dsRNA as the starting material! This way they might have missed some viruses that either do not have a dsRNA replicative phase in their lifecycle or viruses with low replication rates. For example, viruses with circular single-stranded DNA genomes belonging to Genomoviridae do not form dsRNA so using specifically dsRNA as starting material for sequencing might miss out picking such viruses. I think total RNA sequencing or a mixture of total RNA and purified dsRNA fraction as starting material would have been a better choice.
Line 240: Call it a negative-sense/strand RNA virus instead and express its size in “nt” as it is an ssRNA virus.
Line 262: Partitiviruses do have an interrupted poly A tail and it has been reported in several studies.
Viruses belonging to the family are usually tri-segmented. Both genomic and antigenomic strands of negative-sense RNA viruses are encapsidated by nucleocapsids (N) and associated with RdRp to form ribonucleoprotein. They also produce envelop glycoproteins. Fungal negative-sense RNA viruses have also been reported to encode movement proteins as well. However, MP may not be a prerequisite for their replication inside their fungal host. Both GaCLV2 and GaCLV2 encode a putative movement protein. I think CaNSRV-1 here missing some of its genomic segments or it’s a mono-segmented virus. In that case, it would be interesting to study how viruses like CaNSRV-1 or CaNSRV-2 replicate in their fungal hosts given the fact that nucleocapsid structure is a prerequisite (template is the nucleocapsid, not the naked genomic RNA) for –ssRNA virus replication. Also, I am wondering whether the genomes of both viruses have poly-A tails. Please provide this information as well.
Line 465: what about the sequence identity at the terminal regions between RNA1 and RNA2 for CaNSRV-2? I think a figure of terminal sequence alignment would be nice to provide.
Line 545: make it zetapartitiviruses
Line 548: Family names should be italicized.
Author Response
Dear Reviewer 2,
We are grateful to the reviewer for helping us improve our manuscript, ID: viruses-1953051. Those revisions are all valuable and very helpful for revising and improving our paper. Please see below our responses to each of the comments.
>This study for the first time reports mycoviruses in Colletotrichum spp. infecting turfgrass in Japan. In addition, this study reports four new mycoviruses, including one new Alphapartitivirus and Zetapartitivirus, and two new -ssRNA viruses. This article can be accepted after the authors have agreed to alter the manuscript as suggested.
Ans: We seriously considered the reviewers' suggestions and did our best to include them in the manuscript to help it improve further.
>In the summary, check the proper spacing between words carefully.
Ans: We checked the proper spacing between words carefully.
>Italicize Partitiviridaeas it indicates a virus family here.
Ans: We italicized "Partitiviridae.”
>Check throughout the manuscript and wherever you mention an ICTV-recognized virus species, genus or family, italicize the name.
Ans: We have checked carefully and revised.
>Line 23: make it Colletotrichum-associated negative- stranded RNA virus 1 (CaNSRV1)
Ans: We revised as suggested.
>Line 66: it is chestnut blight
Ans: We revised as suggested.
>Line 110: ITS-Internal transcribed spacer
Ans: We revised as suggested.
>In the materials and methods section, the authors said that they collected nine isolates (line 105) but Table1 shows 11 isolates. Why this discrepancy?
Ans: We corrected Table 1.
>Regarding mycovirus sequencing, I am wondering why the authors chose dsRNA as the starting material! This way they might have missed some viruses that either do not have a dsRNA replicative phase in their lifecycle or viruses with low replication rates. For example, viruses with circular single-stranded DNA genomes belonging to Genomoviridae do not form dsRNA so using specifically dsRNA as starting material for sequencing might miss out picking such viruses. I think total RNA sequencing or a mixture of total RNA and purified dsRNA fraction as starting material would have been a better choice.
Ans: We totally agree with the reviewer’s comment that FLDS is unable to detect the whole viromes in the sample. However, our major objective in this study is not to show the advantage of FLDS, nor to reveal the comprehensive virome in the sample. As pointed out by the reviewer, FLDS cannot detect viruses that do not produce dsRNA replicative forms, but it is already proved in a good numbed of individual studies that it can discover wide range of diverse viruses in the hosts(Urayama et al. 2016, Urayama et al. 2018, Chiba et al. 2021, Hirai et al. 2022). For our major objective in this study, to identify and characterize RNA viruses in turfgrass pathogenic Collectotrichum spp., our strategy to use screening by dsRNA electrophoresis and its sequencing by FLDS would be reasonable and sufficient, because dsRNA has been employed as an indicator of mycovirus infection. Moreover, as suggested by previous studies, metatranscriptomic survey of the total RNA by RNAseq could not either solve the complete virome (see the comments to the reviewer 1). Also note that metatranscriptomic analyses can possibly detect some pseudo-viral sequences in the host genome. Sequencing of dsRNA forms by FLDS can evade this problem. To clarify our focus in this study, we modified and deleted some parts from Discussion.
>Line 240: Call it a negative-sense/strand RNA virus instead and express its size in “nt” as it is an ssRNA virus.
Ans: We revised as suggested.
>Line 262: Partitiviruses do have an interrupted poly A tail and it has been reported in several studies.
Ans: We revised the information.
>Viruses belonging to the family are usually tri-segmented. Both genomic and antigenomic strands of negative-sense RNA viruses are encapsidated by nucleocapsids (N) and associated with RdRp to form ribonucleoprotein. They also produce envelop glycoproteins. Fungal negative-sense RNA viruses have also been reported to encode movement proteins as well. However, MP may not be a prerequisite for their replication inside their fungal host. Both GaCLV2 and GaCLV2 encode a putative movement protein. I think CaNSRV-1 here missing some of its genomic segments or it’s a mono-segmented virus. In that case, it would be interesting to study how viruses like CaNSRV-1 or CaNSRV-2 replicate in their fungal hosts given the fact that nucleocapsid structure is a prerequisite (template is the nucleocapsid, not the naked genomic RNA) for –ssRNA virus replication.
Ans: We included suggested information in the manuscript.
>Also, I am wondering whether the genomes of both viruses have poly-A tails. Please provide this information as well.
Ans: We did not detect poly-A tails for the genomes of identified Fungal negative-sense RNA viruses. Information included in the manuscript.
>Line 465: what about the sequence identity at the terminal regions between RNA1 and RNA2 for CaNSRV-2? I think a figure of terminal sequence alignment would be nice to provide.
Ans:
The terminal ends of CaNSRV-2 RNA-1 and RNA-2 were aligned. We found 68% identity between the 5' terminal ends of RNA-1 and RNA-2 and 70% identity between the 3' terminal ends of RNA-1 and RNA-2. A terminal sequence alignment figure has been added as Figure 6(B).
>Line 545: make it zetapartitiviruses
Ans:We corrected as suggested
>Line 548: Family names should be italicized.
Ans: We corrected as suggested
Round 2
Reviewer 1 Report
Unfortunately, the current revision did not revise the introduction section starting at Line 85 where they mentioned the rationale of solely relying on FLDS to characterize the virome. The paragraph is misleading because total RNA sequencing may or may not miss the ends of a viral genome, and it depends on the sequencing depth. This reviewer happens to know many viral contigs are complete using the total RNA sequencing.
If due to budget/lack of personnel/bioinformatics skills and the total RNA sequencing cannot be completed as previously suggested, please include the reasoning in the rebuttal letter in the introduction. However, since total RNA sequencing is very common and doable these days, this reviewer really think that this paper is overall in debt to the readers for the information that could have been reasonably obtained.
Author Response
Dear reviewer 1,
We are grateful to you for helping us improve our manuscript, ID: viruses-1953051. We have addressed your suggestions and comments and have revised the manuscript accordingly.
We agree with you that total RNA sequencing will obtain complete viral contigs if that virus is not new, and there are some complete reference sequences. However, it is almost impossible to define the terminal ends of viral genome in case there are no reference sequences, that is, the targeted virus is new. Thus, we will really appreciate if the reviewer 1 provides us with the example of successful completion of “the unknown viral genome utilizing only total RNA-seq in a reference-free manner”. Indeed, there are several reports in which for the successful completion of the unknown viral genome, total RNA-seq is often augmented with other techniques such as RACE, even in the high-impact Virus Evolution (Sutela et al. 2020).
Reference:
Sutela, S., Forgia, M., Vainio, E.J., Chiapello, M., Daghino, S., Vallino, M., Martino, E., Girlanda, M., Perotto, S. and Turina, M., 2020. The virome from a collection of endomycorrhizal fungi reveals new viral taxa with unprecedented genome organization. Virus evolution, 6(2), p.veaa076.
Please see below our responses to each of the comments from the reviewers.
Reviewer 1
1. Unfortunately, the current revision did not revise the introduction section starting at Line 85 where they mentioned the rationale of solely relying on FLDS to characterize the virome. The paragraph is misleading because total RNA sequencing may or may not miss the ends of a viral genome, and it depends on the sequencing depth. This reviewer happens to know many viral contigs are complete using the total RNA sequencing.
Ans.
Thank you so much for your suggestions. We respect your suggestions and modify the introduction section starting at Line 85. Now the sentence reads as, “Deep-sequencing approaches such as total RNA sequencing, on the other hand, have some limitations. One of these limitations is loss of information on the terminal sequences of a novel or unknown viral genome due to the absence of a reference genome.” We believe it is more effective in this situation to describe the benefits of FLDS by distinguishing between reference-based (known viruses) and reference-free (unknown viruses) scenarios to reviewer 1. In the case of reference-based (known viruses), it is indeed likely that "many viral contigs are complete by employing total RNA sequencing," as the reviewer 1 claimed. However, in the reference-free condition of this study, total RNA sequencing, if performed, cannot be used to obtain, or define, the terminal ends as our focus is unknown RNA viruses. Again, as in our previous response in revision 1, we would want to reaffirm to reviewer 1 that our goal was not to demonstrate that FLDS is superior to total RNA sequencing or to explore the complete virome of the researched fungal isolates. The primary purpose of this work is to identify, characterize, and perform phylogenetic analyses of novel RNA viruses found in turfgrass pathogenic Collectotrichum spp. in Japan. To address this goal, we believe that our strategy of screening mycoviruses using dsRNA electrophoresis and then FLDS is appropriate and adequate, because viruses found in our study is new and we cannot use some reference sequences to define the terminal ends.
- If due to budget/lack of personnel/bioinformatics skills and the total RNA sequencing cannot be completed as previously suggested, please include the reasoning in the rebuttal letter in the introduction. However, since total RNA sequencing is very common and doable these days, this reviewer really think that this paper is overall in debt to the readers for the information that could have been reasonably obtained.
Ans. As stated in revision 1, we would like to confirm the reasons behind employing FLDS in this study rather than total RNA sequencing:
- Successful detection of known and novel RNA viruses from a wide range of organisms or microorganisms using FLDS, an efficient dsRNA targeting metagenomic tool, has already been demonstrated in several independent studies (Urayama et al. 2016, Urayama et al. 2018, Chiba et al. 2021, Hirai et al. 2022) . Moreover, as we stated above, total RNA sequencing cannot define the terminal ends of the virus contig if that virus is novel and there are no reference sequences available.
- The recovery of terminal sequences, uniform read coverage, and an incredibly high recovery rate of viral RNA sequences, including dsRNA and ssRNA viruses, are all features of this FLDS method that have not been previously reported in other metagenomic methods such as total RNA sequencing targeting RNA viruses (Urayama et al. 2016).
- Meanwhile, as suggested by previous studies, even metatranscriptomic survey of the total RNA by RNAseq could not solve the complete virome in true sense. Indeed, the viral read ratio of intracellular RNA viruses in the RNA-seq library is often less than 1% since mRNA and rRNA dominate in the total RNA fraction isolated from biological samples (Matranga et al. 2014), while FLDS enhanced the viral RNA reads by over 300-fold when compared to total RNA-seq (Urayama et al. 2016)
- Our primary goal in this study is not to demonstrate the benefit of FLDS in identifying the ends of viral genomes that produce significant amounts of dsRNA replicative forms. Our main goal in this study is to discover, characterize, and perform phylogenetic analysis on novel RNA viruses that are present in turfgrass pathogenic Collectotrichum spp. in Japan. To address this issue, we believe that our strategy using mycovirus screening by dsRNA electrophoresis and subsequent FLDS is reasonable and sufficient.
References:
Urayama, S.I., Takaki, Y. and Nunoura, T., 2016. FLDS: a comprehensive dsRNA sequencing method for intracellular RNA virus surveillance. Microbes and environments, p.ME15171.
Urayama, S.I., Takaki, Y., Nishi, S., Yoshida‐Takashima, Y., Deguchi, S., Takai, K. and Nunoura, T., 2018. Unveiling the RNA virosphere associated with marine microorganisms. Molecular ecology resources, 18(6), pp.1444-1455.
Chiba, Y., Oiki, S., Yaguchi, T., Urayama, S.I. and Hagiwara, D., 2021. Discovery of divided RdRp sequences and a hitherto unknown genomic complexity in fungal viruses. Virus evolution, 7(1), p.veaa101.
Hirai, J., Urayama, S.I., Takaki, Y., Hirai, M., Nagasaki, K. and Nunoura, T., 2022. RNA Virosphere in a Marine Zooplankton Community in the Subtropical Western North Pacific. Microbes and environments, 37(5), p.ME21066.
Matranga, C.B., K.G. Andersen, S. Winnicki, et al. 2014. Enhanced methods for unbiased deep sequencing of Lassa and Ebola RNA viruses from clinical and biological samples. Genome Biol. 15:519.